# High-Intensity Interval Training Reduces Liver Enzyme Levels and Improves MASLD-Related Biomarkers in Overweight/Obese Girls

**DOI:** 10.3390/nu17010164

**Published:** 2025-01-01

**Authors:** Wissal Abassi, Nejmeddine Ouerghi, Mohamed Bessem Hammami, Nidhal Jebabli, Moncef Feki, Anissa Bouassida, Katja Weiss, Beat Knechtle

**Affiliations:** 1Research Unit “Sport Sciences, Health and Movement” (UR22JS01), High Institute of Sport and Physical Education of Kef, University of Jendouba, Kef 7100, Tunisia; wissalabassi93@gmail.com (W.A.); najm_ouerghi@hotmail.com (N.O.); jnidhal@gmail.com (N.J.); bouassida_anissa@yahoo.fr (A.B.); 2Faculty of Medicine of Tunis, Rabta Hospital, LR99ES11, University of Tunis El Manar, Tunis 1007, Tunisia; bessem_hammami@yahoo.fr (M.B.H.); monssef.feki@gmail.com (M.F.); 3High Institute of Sport and Physical Education of Gafsa, University of Gafsa, Gafsa 2100, Tunisia; 4Institute of Primary Care, University of Zurich, 8091 Zurich, Switzerland; katja@weiss.co.com; 5Medbase St. Gallen Am Vadianplatz, 9000 St. Gallen, Switzerland

**Keywords:** insulin resistance, liver enzyme, nonalcoholic fatty liver disease, obesity, physical training, lipid profile

## Abstract

Background/Objectives: Despite the abundant body of evidence linking high-intensity interval training (HIIT) to cardiometabolic markers, little is known about how HIIT affects liver enzymes, particularly in obese adolescents. This study aimed to investigate the effects of HIIT on metabolic dysfunction-associated steatotic liver disease (MASLD)-related biomarkers in overweight/obese adolescent girls. Methods: Thirty-three overweight/obese adolescent girls (age, 17.0 ± 1.15 yr.; body mass index, 33.3 ± 4.77 kg/m^2^) were randomly assigned to HIIT (*n* = 17) or control (*n* = 16) groups. The HIIT group participated in a nine-week HIIT program (three times weekly) without caloric restriction. Maximal aerobic speed, body composition indexes, blood pressure, MASLD-related biomarkers [liver enzymes (alanine aminotransferase (ALT) and aspartate aminotransferase (AST)), plasma lipids, uric acid, platelet count, and homeostasis model assessment index for insulin-resistance (HOMA-IR)] were examined at baseline and after the intervention. Results: Significant “time × group” interactions were found for body composition indexes, systolic blood pressure, maximal aerobic speed, liver enzymes ALT and AST, plasma lipids, glucose, and HOMA-IR. The HIIT program resulted in an increase in maximal aerobic speed (*p* = 0.035) and a decrease in body composition and plasma lipids (*p* < 0.01), systolic blood pressure (*p* = 0.011), ALT (*p* = 0.013), AST (*p* = 0.012), and HOMA-IR (*p* = 0.01), but no significant changes in uric acid and platelet count. None of these markers changed in the control group. Conclusions: HIIT resulted in an improvement in MASLD-related biomarkers. HIIT could be an effective exercise therapy to prevent and reverse MASLD in adolescents with obesity.

## 1. Introduction

Metabolic dysfunction-associated steatotic liver disease (MASLD), formerly known as nonalcoholic fatty liver disease, is characterized by the presence of hepatic steatosis with the onset of metabolic risk factors, most notably type 2 diabetes mellitus and overweight [1]. The prevalence of MASLD increases from 25% in the general population to 60% in high-risk populations, such as obese and sedentary individuals [2].

MASLD is the most frequent cause of chronic liver disease in children and adolescents with obesity [3,4]. Pediatric MASLD has dramatically increased with the epidemic of overweight and obesity; MASLD prevalence ranges from 5% and 10% in the general pediatric population [3] to 40% among children and adolescents with obesity [5,6,7]. MASLD is commonly considered as the liver-related manifestation of the metabolic syndrome and is highly associated (50–80%) with a spectrum of other metabolic comorbidities, such as insulin resistance, dyslipidemia, hypertriglyceridemia, and hypertension [8]. Pediatric-onset MASLD presents with an aggressive progression and a higher incidence in adulthood that can lead to the development of the inflammatory subtype of metabolic dysfunction-associated steatohepatitis (MASH), which can further progress to various stages of liver fibrosis, cirrhosis, and ultimately to hepatocellular carcinoma [9,10].

Increasing attention has been paid to accurate early diagnosis to avert disease severity and monitor the disease progression over time. Liver enzymes, mainly alanine aminotransferase (ALT) and aspartate aminotransferase (AST), are non-invasive and inexpensive biomarkers commonly used for MASLD screening [2], particularly in children and adolescents with obesity [3,11,12]. Serum lipids and uric acid concentrations, insulin resistance indexes, platelet count (PC), and mean platelet volume (MPV) can also be used as surrogate biomarkers for MASLD in the pediatric population [13,14,15,16].

The main approach to managing MASLD and associated metabolic disorders is based on regular physical exercise and lifestyle changes [7,10,17]. There is a consensus in the literature that various exercise modalities and intensities generally benefit MASLD [10,18,19]. Moderate-intensity aerobic exercise has been shown to reduce hepatic steatosis by approximately 2–4% in adults with metabolic-associated fatty liver disease (MAFLD) [18]. Additionally, low-intensity resistance exercise elicits beneficial effects in combating MASLD, particularly by decreasing insulin resistance and improving metabolic flexibility as potential mechanisms to mitigate liver injury [10]. Furthermore, moderate-to-high-intensity aerobic, resistance, and combined exercises have been demonstrated to reduce liver fat and improve liver enzymes and HOMA-IR [19].

However, these types of training are time-consuming, which represents a constraint for youths who spend long hours on schoolwork, playing with electronic devices, and connecting to the internet. A time-efficient alternative to aerobic exercise is high-intensity interval training (HIIT) [20,21]. HIIT has been proven to improve body composition [22,23], plasma lipids [21], and insulin sensitivity in overweight/obese youths [21,24]. Despite the abundant evidence for an association between HIIT and cardiometabolic markers, little is known about changes in liver enzymes induced by HIIT [25], especially in obese adolescents. Moreover, data on the effect of HIIT on other MASLD-related biomarkers, such as uric acid and platelet indexes in obese children/adolescents, are missing.

The current study aimed to evaluate the effects of HIIT on selected MASLD-related biomarkers in adolescents with overweight/obesity. We hypothesized that the HIIT program contributes to reduce liver enzyme levels and improve body composition, circulating lipids, and insulin sensitivity.

## 2. Materials and Methods

### 2.1. Participants

G*Power software (version 3.1, Germany) was used to estimate the sample size. Using a partial effect size of 0.55, with a power of 0.95 at an alpha level of 0.05 (two groups and two measurements), we obtained a sample size of 14 participants for each group. Overweight/obese girls from secondary schools in the Kef region (Tunisia) were invited to participate in the study. Consenting girls aged between 15 and 18 years, with a BMI ≥ 95th percentile for age and no contraindication to physical activity, were eligible for inclusion. Among 49 girls who were willing to participate, 11 did not meet eligibility criteria, and 2 subsequently declined due to logistical issues. Thirty-six girls were included and randomly distributed between the training (HIITG, *n* = 18) and control (CG, *n* = 18) groups. During the intervention program, no injuries occurred, but three girls did not complete the training program for personal reasons and thus were excluded from the analysis. Finally, 33 girls, 17 in the HIIT group and 16 in the control group, completed the training program (Figure 1). The participants’ usual physical activity consisted of weekly two-hour physical education lessons. This usual physical activity, as well as habitual eating behaviors and tasks, were maintained over the intervention period in all participants. The study was conducted following the guidelines of the Declaration of Helsinki. The protocol was approved by the local Ethics Committee of the High Institute of Sports and Physical Education of Kef (approval code ISSEP-04/2019), and written informed consent was obtained from the girls’ parents or legal guardians.

### 2.2. Study Protocol

A nine-week randomized trial was conducted from March to May 2019, with temperature varying from 14 °C to 19 °C and humidity ranging between 71% and 64%. Body composition, maximal aerobic speed (MAS), and biochemical parameters were assessed in all participants before the beginning and after the end of the training program. Body mass (BM) and body fat percentage (BF) were obtained using standard conventional methods (in light clothing and without shoes) with a TANITA scale (Tanita BC-533, Tokyo, Japan). Height was measured using a stadiometer. Body mass index (BMI) was calculated from body mass divided by square height. Waist circumference (WC) was obtained with a non-deformable tape measure between the lower rib margin and the iliac crest. Systolic (SBP) and diastolic (DBP) blood pressures were determined using an automatic blood pressure monitor (Omron BP652, Omron Healthcare Inc., Vernon Hills, IL, USA) after 10 min of relaxation in a seated position. Maximal aerobic speed (MAS) was determined by a graded exercise test until exhaustion, the “Spartacus test” as previously described [23], and maximal heart rate (HRmax) was recorded using a heart rate monitor (S810, Polar, Kempele, Finland) during this incremental running test.

### 2.3. Training Program

Each training session was preceded by a 15 min warm-up and cool-down at 50% MAS. The warm-up session included moderate-intensity jogging (5 min), dynamic stretching (5 min), and acceleration running (5 min). The training program was carried out for 9 consecutive weeks with 3 sessions per week [23,24]. Specifically, the protocol consisted of two series of 6 repetitions of 30 s work at 100–105% of MAS and 30 s active recovery at 50% of MAS. Participants were required to increase the number of repetitions from the 4th week during the intervention. The intensity to be achieved was increased from the 7th week of the training program (Table 1). Girls in the control group continued their regular physical education classes but did not participate in additional training programs.

### 2.4. Blood Sampling and Analysis

Following a 12 h fast, venous samples were taken around 8 a.m. one day before all participants completed pre-intervention venipuncture and two days post-intervention venipuncture performed by a trained phlebotomist. All venous blood samples were centrifuged at 2000× *g* for 25 min and plasma was frozen at −40 °C until analysis (within 3 months). The liver enzyme (i.e., ALT and AST), total cholesterol (TC), HDL cholesterol (HDL-C), triglyceride (TG), glucose, and uric acid were analyzed using an automated immunoassay system (AU480 Chemistry Analyzer; Beckman Coulter, Brea, CA, USA). LDL cholesterol (LDL-C) was calculated using the Friedwald formula [26]. Platelet count (PC) and mean platelet volume (MPV) were determined using an automated cell counter (XN450; Sysmex, Norderstedt, Germany). Plasma insulin was measured by radioimmunoassay kits (Beckman Coulter Company, Paris, France). Insulin resistance was evaluated by the homeostatic model assessment for insulin resistance (HOMA-IR) as follows: HOMA-IR = [fasting insulin (μU/mL)] × fasting glucose (mmol/L)]/22.5 [27].

### 2.5. Statistical Analysis

Data were analyzed using SPSS software (Version 21.0; IBM Corp., Armonk, NY, USA). The Kolmogorov–Smirnov test was used to confirm data distribution normality. The homogeneity of variance was tested with the F-test. A two-way mixed analysis of variance (ANOVA) with repeated measures (2 groups × 2 time points: pre- and post-intervention) was performed to test the interaction effect for the group by time on the outcome variables. When a significant interaction existed, paired *t*-tests with Bonferroni correction were performed to examine the difference in pre- and post-measures in each group. Independent *t*-tests were performed to test the differences between the two groups. Effect size statistics (ES) were established based on Cohen’s classification as small (0.00 < d < 0.49), medium (0.50 < d < 0.79), and large (d < 0.80) [28]. Data were expressed as mean ± standard deviation (SD), and *p*-values of <0.05 were considered significant.

## 3. Results

Pre- and post-training values and “time × group” interaction for all variables analyzed are reported in Table 2. At baseline, there were no significant differences between the two groups for all the variables. A significant “time × group” interaction was found for all anthropometric parameters, SBP, MAS, and HRmax. Intragroup analyses showed a significant improvement in body mass (*p* < 0.001, ES = 0.25), BMI (*p* = 0.001; ES = 0.29), BF (*p* < 0.001, ES = 0.52), WC (*p* = 0.003, ES = 0.37), SBP (*p* = 0.011, ES = 0.80), MAS (*p* = 0.035, ES = 0.54), and HRmax (*p* = 0.035, ES = 0.73) after the 9-week intervention in the HIITG. However, no significant changes were observed for all the variables in CG. Between-group analyses demonstrated higher MAS (*p* = 0.010, ES = 0.99) post-intervention but lower HRmax (*p* < 0.001, ES = 1.40) in the training group. A significant “time × group” interaction was also found for biochemical markers, including ALT, AST, plasma lipids, glucose, insulin, and HOMA-IR. Intragroup analysis showed a significant decrease in plasma TC (*p* = 0.003, ES = 0.45), triglyceride (*p* = 0.005, ES = 0.54), LDL-C (*p* = 0.002, ES = 0.60), ALT (*p* = 0.013, ES = 0.47), AST (*p* = 0.012, ES = 0.58), and HOMA-IR (*p* = 0.010, ES = 0.47), and a significant increase in plasma HDL-C (*p* = 0.013, ES = 0.36) after the 9-week intervention program in the HIITG. The between-group comparison showed lower plasma AST (*p* = 0.036; ES = 0.79) post-intervention in the training group. No significant changes were observed for all biochemical markers in CG. Figure 2 shows the anthropometric, physical, and laboratory variable changes in the training and control groups.

## 4. Discussion

This study evaluated the effect of a nine-week HIIT program on MASLD-related biomarkers in adolescent females with overweight/obesity. The main findings were (1) a significant decrease in liver enzymes ALT and AST, (2) an improvement in cardiometabolic biomarkers (body composition, plasma lipids, insulin sensitivity, and blood pressure), and (3) no change in platelet indexes and uric acid levels. These findings suggest that HIIT may protect against MASLD.

Childhood obesity has become a prominent public health concern in transitional countries like Tunisia. The country is experiencing socioeconomic and nutritional transition and urbanization. Traditional Mediterranean diets have been supplanted by high-energy, fatty, and salty diets [29,30]. Concurrently, children and adolescents have become more engaged in sedentary activities such as surfing the internet and playing with electronic devices, drastically reducing physical activity [30,31]. The prevalence of overweight among Tunisian adolescents rose from 10.8% in 1996 to 18.9% in 2005, then 30.5% in 2016 [32]. A recent study involving 1399 Tunisian high-school students showed that 20.4% are overweight and 7% are obese [33].

MASLD is strongly linked to pediatric obesity [34]. BMI and body fat are key determinants of MASLD in obese children and adolescents [12,35]. In the present study, the HIIT program decreased BMI, BF, and WC, corroborating previous data [22,23]. It was shown that a moderate reduction in body mass and BMI in overweight/obese children improves liver function and decreases ALT levels [36]. A weight loss of <5.0% was shown to induce hepatic benefit [37]. In the present study, the training program decreased body mass by 3%, which suggests that HIIT is expected to improve liver health. Weight loss may be explained by exercise-induced fat oxidation via the activation of mitochondrial beta-hydroxy acyl-CoA dehydrogenase and glycogen resynthesis [38].

A nine-week HIIT program resulted in a decrease in circulating TC by 7%, LDL-C by 12%, and TG by 6%, and an increase in HDL-C levels by 7% in these overweight/obese girls. This finding agrees with the literature data on the beneficial effects of HIIT on circulating lipids [21]. Dyslipidemia is an important risk factor for the development of MASLD in children and adolescents with obesity [39,40]. By improving circulating lipid status, HIIT programs may reduce the risk of MASLD.

Insulin resistance (IR) is recognized as a common pathophysiological abnormality underlying obesity, dyslipidemia, and MASLD [40,41,42]. The increase in visceral adipose tissue mass determines an increase in free fatty acid (FFA) levels, which induce inhibition of the insulin signaling pathways. IR causes inhibition of adipose tissue lipolysis via the regulation of transcription factors, including SREBP-1, ChREBP, and PPAR-γ [42]. It also stimulates liver lipogenesis, accumulating triglycerides within the hepatocytes and leading to MASLD [43]. This condition can itself promote IR, establishing a vicious cycle. Therefore, it is likely that improvement in insulin sensitivity will help to reduce MASLD progression and this pathway could be considered a key therapeutic target. The current study showed that HIIT resulted in significant decreases in fasting glucose, insulin, and HOMA-IR, which indicates an improvement in insulin sensitivity in these overweight/obese adolescents. Exercise-induced activation of the MAPK pathway may also contribute to insulin sensitivity improvement [44].

The HIIT program resulted in an improvement in MAS, HRmax, and blood pressure. Numerous studies have shown the ability of HIIT to improve cardiorespiratory fitness in overweight and obese youth [21,22,23,24]. The increase in maximal oxygen uptake and cardiac output by HIIT may explain aerobic fitness improvement [45]. The development of obesity increases the likelihood of hypertension in childhood [46], which is closely associated with MASLD development [37,47]. In the current study, HIIT intervention has significantly decreased SBP (~2%). Taylor et al. [48] showed that a decrease in SBP above 4 mm Hg reduces cardiovascular disease mortality by 5~20%. Therefore, HIIT intervention may significantly reduce cardiovascular problems in obese adolescents. It has become evident that WC is positively associated with SBP in pediatric obesity [49], which is consistent with the findings of the present study; both WC (~3%) and SBP (~2%) decreased in the HIIT group.

MASLD covers a spectrum of liver diseases that can range from simple fatty liver to MASH, cirrhosis, and hepatocellular carcinoma [9,10]. Physical training intervention is a therapeutic strategy against MASLD and MASH [10,18,19,50]. Exercise training decreased fibrosis, apoptosis, and oxidative stress [51,52]. Aerobic and resistance interventions have been shown to elicit reductions in liver fat [53]. Furthermore, HIIT lessens liver fat [54] and hepatic inflammation [55].

The liver enzymes AST, ALT, and gamma-glutamyl transferase are closely related to liver fat accumulation and the development of MASLD [37,56]. In this study, we found that the HIIT program is effective in reducing plasma ALT and AST levels. Few studies have investigated the effects of HIIT on liver enzymes [57,58,59]. Liver enzyme levels were reduced after 12 weeks of HIIT in obese individuals [57] and adult females without obesity [25]. However, other studies showed no change in ALT or AST following a 6-week HIIT program in healthy young individuals [59] or a 12-week HIIT intervention in patients with type 2 diabetes [58]. The mechanisms explaining liver enzyme reduction after HIIT have been poorly investigated. It may be related to HIIT’s effect on intrahepatic lipid reduction [25], which could be influenced by blood lipid and body mass modifications.

Hyperuricemia is increasing in obese children and adolescents, which raises the risk of MASLD [60,61]. Our findings demonstrated a non-significant reduction in uric acid levels (~4%). To date, reports on the effect of training on uric acid in obese people are scarce. Most available studies focus on uric acid responses following lifestyle modification encompassing diet and physical activity [62]. A significant reduction in uric acid after a 3-month intervention (training + diet) in overweight/obese children and adolescents was confirmed [62]. Lamina et al. showed that 8 weeks of interval training without diet intervention significantly decreased uric acid levels [63]. Given the established close relationship between uric acid and hypertension, we can speculate that the response of uric acid to HIIT could be explained by blood pressure reduction.

There is a lack of reports on the effect of HIIT on platelet count. While data from platelets’ response to exercise is inconclusive, most studies indicated a transient increase in platelet count after different exercise protocols [64]. Others demonstrated no changes in platelet count after exercise [65]. The current study showed no platelet index (PC and MPV) changes. Discrepancies may originate from differences in exercise protocols (e.g., intensity, duration) and the populations investigated (e.g., age, gender, body composition, inflammatory status).

HIIT could be integrated into school curricula and clinical settings for overweight/obese adolescents to reduce body mass excess and liver fat, improve physical fitness, and promote cardiometabolic health. This would prevent/reverse MASLD and avoid its awful complications. Further research is needed to clarify the mechanisms underlying the effects of HIIT on fatty liver while focusing on the potential direct action of muscle metabolism on the liver.

The study was not without limitations. First, the number of subjects in each group was relatively small, which may have influenced the statistical power to detect differences between groups. Second, only girls were included in this study, which meant that we could not better explore gender differences in the effect of exercise intervention. Third, the study involved overweight/obese adolescents, and therefore it is difficult to generalize the findings to normal-weight subjects of different ages. Additionally, we did not assess hormones produced during puberty that may influence our results. Puberty is associated with hormonal changes, mainly the release of sex hormones (estradiol/progesterone). Also, prolactin, growth hormone, insulin, and insulin-like growth factor-1 levels rise noticeably. These changes are responsible for fat mass gains that alter aerobic capacity and cardiometabolic parameters by increasing body size and the dimensions of the heart, lungs, muscles, and circulatory system [66,67]. Furthermore, the current study did not control and adjust for dietary and energy intake. These factors might significantly influence the investigated biomarkers. Moreover, the quality of the study could be further improved by recording girls’ adherence to exercise training. Indeed, numerous studies have suggested that obese adolescents have some difficulties in participating in HIIT programs [68]. Finally, the study did not include measurement of liver fat by DEXA or CT, nor additional biochemical biomarkers of liver damage. Therefore, confirmation of the diagnosis of MASLD was lacking in these participants. However, most girls showed higher concentrations of ALT within the reference range, which was proven to predict MASLD [56]. Despite this, the study showed, without a doubt, improvements in several MASLD-related biomarkers, suggesting a likely beneficial effect of HIIT on MASLD and associated morbidities. However, to prevent MASLD, not only exercise therapy but also improvements in diet and physical habits should be implemented at the same time.

## 5. Conclusions

The study showed that a nine-week HIIT program improves body composition and reduces liver enzymes, plasma lipids, blood pressure, and insulin resistance in overweight/obese adolescent girls. These findings suggest that HIIT is effective in fighting MASLD and its complications.

## Figures and Tables

**Figure 1 nutrients-17-00164-f001:**
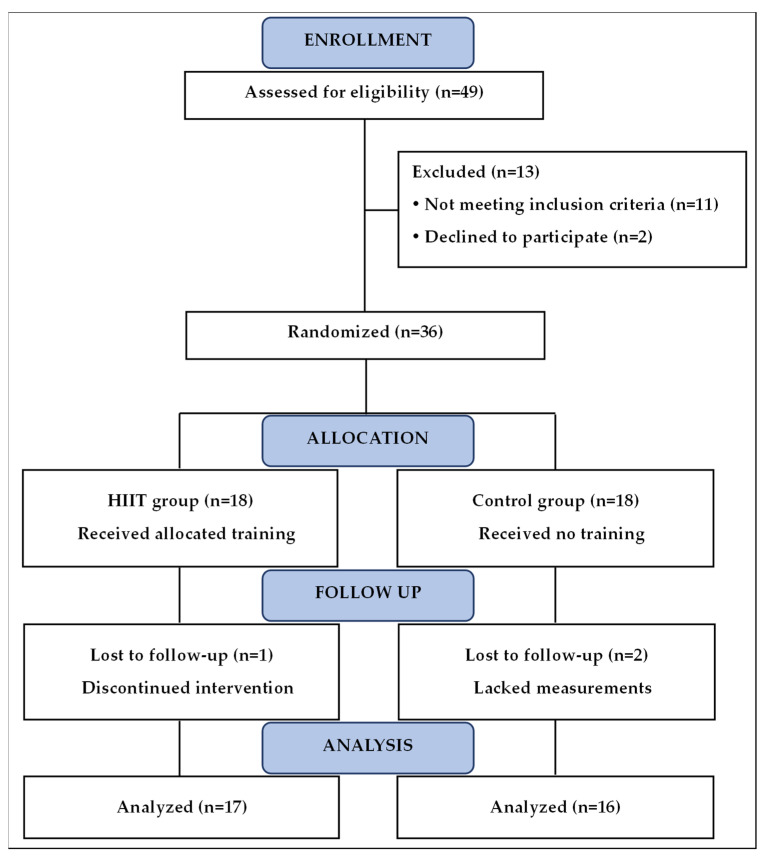
Flowchart of study. HIIT, high-intensity interval training.

**Figure 2 nutrients-17-00164-f002:**
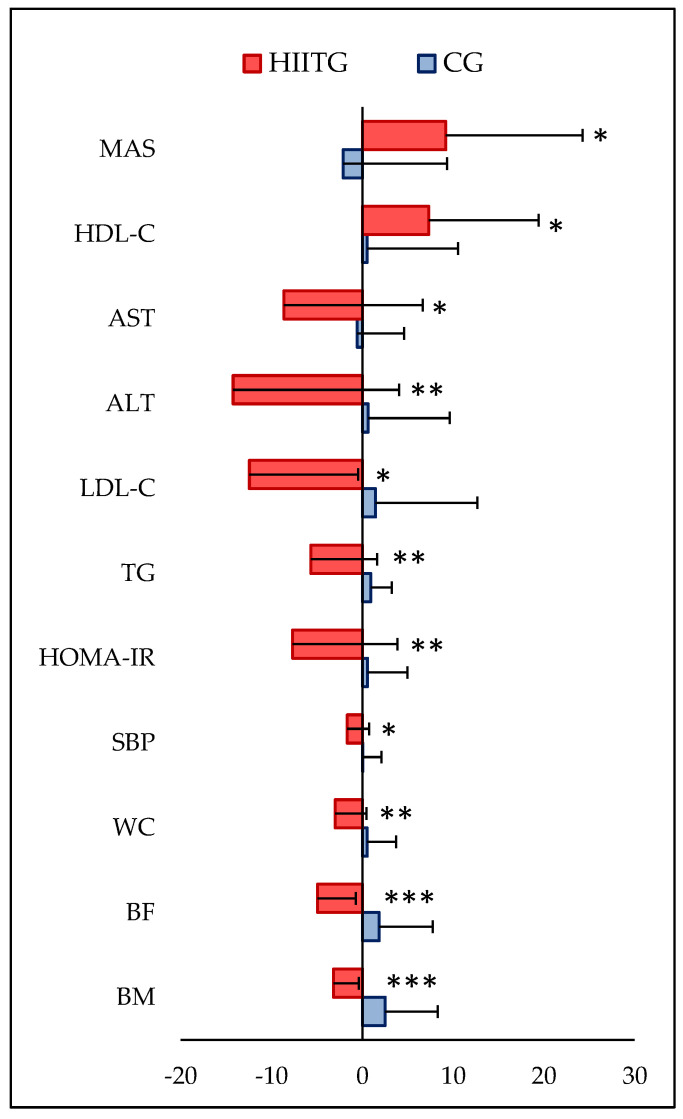
Changes in anthropometric, biochemical, and physical variables following a nine-week intervention program in high-intensity interval training (HIIT) and control (CG) groups. ALT, alanine aminotransferase; AST, aspartate aminotransferase; BM, body mass; BF, body fat percentage; HDL-C, HDL-cholesterol; HOMA-IR, homeostatic model assessment for insulin resistance; LDL-C, LDL-cholesterol; MAS, maximal aerobic speed; SBP, systolic blood pressure; TG, triglyceride; WC, waist circumference. *, *p* < 0.05; **, *p* < 0.01; ***, *p* < 0.001 (compared to control group; *t*-test for independent samples).

**Table 1 nutrients-17-00164-t001:** The training program details.

	Weeks 1 to 3	Weeks 4 to 6	Weeks 7 to 9
Work: Rest duration, s	30:30	30:30	30:30
Work: Rest intensity	100:50% of MAS	100:50% of MAS	105:50% of MAS
Number of repetitions	6	8	8
Number of sets	2	2	2
Rest between sets, s	240	240	240

MAS: maximal aerobic speed.

**Table 2 nutrients-17-00164-t002:** Pre- and post-intervention anthropometric, physical, and laboratory variables and (time × group) interaction in the training and control groups.

	Control Group (*n* = 16)		Training Group (*n* = 17)		*p*-Value		
	Pre	Post	Pre	Post	P1	P2	P3
Age (year)	17.3 ± 1.10		16.7 ± 1.32				
Height (m)	1.59 ± 0.08		1.60 ± 0.07				
Body mass (Kg)	84.0 ± 9.74	85.9 ± 9.24	85.1 ± 11.4	82.3 ± 11.0	0.782	0.001	0.001
Body mass index (Kg/m^2^)	33.6 ± 5.45	34.3 ± 4.83	33.1 ± 4.18	32.0 ± 3.81	0.765	0.001	0.001
Body fat (%)	32.7 ± 2.76	33.3 ± 2.88	34.5 ± 3.87	32.7 ± 3.16	0.149	0.001	0.001
Waist circumference (cm)	101 ± 8.47	101 ± 7.82	107 ± 10.4	104 ± 8.34	0.061	0.003	0.007
Systolic blood pressure (mm Hg)	125 ± 3.40	125 ± 3.91	125 ± 2.74	123 ± 2.71	0.809	0.011	0.034
Diastolic blood pressure (mm Hg)	77.4 ± 3.18	77.3 ± 3.76	78.1 ± 4.19	77.0 ± 3.88	0.636	0.015	0.237
Heart rate max (bpm)	198 ± 3.18	198 ± 1.43	198 ± 2.50	196 ± 2.34	0.575	0.035	0.048
Maximal aerobic speed (km/h)	10.8 ± 1.24	10.4 ± 1.03	10.9 ± 1.64	11.8 ± 1.74	0.709	0.035	0.022
Alanine aminotransferase (UI/L)	19.1 ± 8.45	19.4 ± 9.12	18.8 ± 5.37	16.2 ± 6.24	0.923	0.013	0.008
Aspartate aminotransferase (UI/L)	22.6 ± 3.36	22.8 ± 3.53	22.5 ± 5.01	20.1 ± 3.51	0.918	0.012	0.010
Total cholesterol (mg/dL)	179 ± 23.8	181 ± 38.2	177 ± 37.1	163 ± 24.5	0.897	0.003	0.035
Triglycerides (mg/dL)	115 ± 10.4	115 ± 9.28	118 ± 14.2	111 ± 12.9	0.432	0.005	0.002
HDL-cholesterol (mg/dL)	44.9 ± 6.60	44.5 ± 6.42	43.7 ± 8.20	46.7 ± 8.64	0.656	0.013	0.031
LDL-cholesterol (mg/dL)	111 ± 17.5	113 ± 33.6	110 ± 30.5	94.7 ± 21.1	0.911	0.002	0.026
Fasting glucose (mmol/L)	4.83 ± 0.32	4.85 ± 0.37	4.82 ± 0.46	4.60 ± 0.59	0.982	0.041	0.030
Fasting insulin (µU/mL)	18.3 ± 1.47	18.3 ± 1.38	18.1 ± 1.94	17.4 ± 1.50	0.687	0.030	0.044
HOMA-IR	3.94 ± 0.55	3.96 ± 0.56	3.90 ± 0.76	3.57 ± 0.70	0.882	0.010	0.008
Uric acid (µmol/L)	268 ± 47.5	270 ± 43.5	280 ± 63.8	268 ± 67.6	0.523	0.083	0.094
Platelet count (10^3^/mm^3^)	290 ± 89.4	285 ± 87.1	292 ± 61.7	294 ± 65.5	0.940	0.807	0.731
Mean platelet volume (fl)	10.0 ± 0.89	10.4 ± 0.67	10.6 ± 1.33	10.5 ± 0.97	0.147	0.542	0.151

Data are expressed as mean ± SD; P1, the *p*-value for pre-intervention differences between the training and control groups; P2, the *p*-value for the differences between pre- and post-intervention in the training group; P3, the *p*-value for the interaction (time × group).

## Data Availability

The data presented in this study is available at the request of the corresponding author for privacy reasons.

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
