# Peer review of "High-Intensity Interval Training Reduces Liver Enzyme Levels and Improves MASLD-Related Biomarkers in Overweight/Obese Girls"

_nutrients, 2025, doi:10.3390/nu17010164_

Round 1
Reviewer 1 Report
Comments and Suggestions for Authors
First of all, I think this is a very relevant topic. As the authors indicate, there is a lot of research surrounding HIIT, but we are still in an early phase, especially considering the implications this type of physical activity has on physiological indicators, as well as on epigenetics (although this is not the focus of the current study). HIIT training has great potential, not only for improving physical fitness but also because evidence suggests it alters genetic expression. Or, as in this case, it modifies indicators that reflect improvements in health.
The study is very well designed, with a very clear flow chart. However, it is important to note that the sample size is very small, as the authors themselves acknowledge in lines 310-311.
One suggestion for the authors is to reorganize Table 2. They should present more clearly three indicators: (1) differences between groups, (2) differences before vs after, and (3) interaction. I suggest the following: in the last columns of the table, remove the F value and report the p-value for the differences between groups, before vs after, and for the interaction. Three columns with the p-values.
Finally, in the conclusions, all references to implications should be removed (this should be moved to the discussion). In the conclusions, I recommend "getting to the point"; the first sentence is enough.
Author Response
Reviewer 1
First of all, I think this is a very relevant topic. As the authors indicate, there is a lot of research surrounding HIIT, but we are still in an early phase, especially considering the implications this type of physical activity has on physiological indicators, as well as on epigenetics (although this is not the focus of the current study). HIIT training has great potential, not only for improving physical fitness but also because evidence suggests it alters genetic expression. Or, as in this case, it modifies indicators that reflect improvements in health.
The study is very well designed, with a very clear flow chart. However, it is important to note that the sample size is very small, as the authors themselves acknowledge in lines 310-311.
One suggestion for the authors is to reorganize Table 2. They should present more clearly three indicators: (1) differences between groups, (2) differences before vs after, and (3) interaction. I suggest the following: in the last columns of the table, remove the F value and report the p-value for the differences between groups, before vs after, and for the interaction. Three columns with the p-values.
Answer: Thank you for your pertinent comment. Table 2 was revised as suggested, and the three p-values were displayed (please see Table 2, lines 237 to 241 in the revised manuscript).
Finally, in the conclusions, all references to implications should be removed (this should be moved to the discussion). In the conclusions, I recommend "getting to the point"; the first sentence is enough.
Answer: The conclusion was revised according to your recommendation (lines 348 to 351 in the revised manuscript.

Reviewer 2 Report
Comments and Suggestions for Authors
In this article, Wissal Abassi and colleagues evaluated the effects of HIIT on NAFLD-related biomarkers in adolescents with overweight/obesity. They showed that HIIT program contributes to reducing liver enzyme levels and improving body composition, circulating lipids, and insulin sensitivity. The article is very interesting, and the rationale behind the study is well-founded. However, I have a few comments for the authors that could enhance the robustness of the article and make it suitable for publication in Nutrients. In summary, MASLD is seen as a more accurate and inclusive term than NAFLD because it ties liver disease to its metabolic causes rather than merely excluding alcohol consumption as the primary cause. I advise the authors to strictly replace the term NAFLD with MASLD. Furthermore, the authors should update the bibliography with more recent articles emphasising MAFLD/MASLD. Exercise and lifestyle changes are considered effective therapeutic approaches for the treatment of MASLD and associated metabolic disorders (https://pubmed.ncbi.nlm.nih.gov/39229589/; https://pubmed.ncbi.nlm.nih.gov/35334826/). However, there are various types of exercise and different levels of intensity. The authors should expand on this point by analyzing recently published articles. In the discussion section, the authors should clarify the potential limitations of the study, particularly in only recruiting overweight/obese adolescents. The influence of hormones produced during puberty should also be considered. Could high-intensity interval training have practical applications and be included in school curricula or clinical settings for adolescents with obesity or overweight status?
Author Response
Reviewer 2
In this article, Wissal Abassi and colleagues evaluated the effects of HIIT on NAFLD-related biomarkers in adolescents with overweight/obesity. They showed that HIIT program contributes to reducing liver enzyme levels and improving body composition, circulating lipids, and insulin sensitivity. The article is very interesting, and the rationale behind the study is well-founded. However, I have a few comments for the authors that could enhance the robustness of the article and make it suitable for publication in Nutrients. In summary, MASLD is seen as a more accurate and inclusive term than NAFLD because it ties liver disease to its metabolic causes rather than merely excluding alcohol consumption as the primary cause. I advise the authors to strictly replace the term NAFLD with MASLD. Furthermore, the authors should update the bibliography with more recent articles emphasizing MAFLD/MASLD.
Answer: Thank you for your interesting comment. We agree with the expert reviewer and more information has been added in the revised manuscript. The term “NAFLD” was changed to “MASLD” and “NASH” to “MASH” through the manuscript. The following paragraph has been added in the introduction section" “Metabolic dysfunction-associated steatotic liver disease (MASLD), formerly known as nonalcoholic fatty liver disease, is characterized by the presence of hepatic steatosis with the onset of metabolic risk factors, most notably type 2 diabetes mellitus and overweight [1]. The prevalence of MASLD increases from 25% in the general population to 60% in high-risk populations, such as obese individuals [2]. The prevalence increases with obesity and lack of physical activity [2].
MASLD is the most frequent cause of chronic liver disease in children and ado-lescents with obesity [3,4]. Pediatric MASLD has dramatically increased with the epi-demic of overweight and obesity, MASLD prevalence ranges from 5% and 10% in the general pediatric population [3], to 40% among children and adolescents with obesity [5-7]. MASLD is commonly considered as the liver-related manifestation of the meta-bolic syndrome and is highly associated (50%–80%) with a spectrum of other metabolic comorbidities, such as insulin resistance, dyslipidemia, hypertriglyceridemia, and hy-pertension [8]. Pediatric-onset MASLD presents with an aggressive progression and a higher incidence in adulthood that can lead to the development of the inflammatory subtype of metabolic dysfunction-associated steatohepatitis (MASH), which can further progress to various stages of liver fibrosis, cirrhosis, and ultimately to hepatocellular carcinoma [9,10].” (lines 42 to 57, Introduction section).
The reference list has been updated with more recent articles which have a focus on MAFLD/MASLD.
Exercise and lifestyle changes are considered effective therapeutic approaches for the treatment of MASLD and associated metabolic disorders (https://pubmed.ncbi.nlm.nih.gov/39229589/; https://pubmed.ncbi.nlm.nih.gov/35334826/). However, there are various types of exercise and different levels of intensity. The authors should expand on this point by analyzing recently published articles.
Answer: Thank you for your interesting comment. We agree with the expert reviewer and more information has been added in the revised manuscript “The main approach to managing MASLD and associated metabolic disorders is based on regular physical exercise and lifestyle changes [7,10,17]. There is a consensus in the literature that various exercise modalities and intensities generally benefit MASLD [10,18,19]. Moderate-intensity aerobic exercise has been shown to reduce hepatic stea-tosis by approximately 2–4% in adults with metabolic-associated fatty liver disease (MAFLD) [18]. Additionally, low-intensity resistance exercise elicits beneficial effects in combating MASLD, particularly by decreasing insulin resistance and improving met-abolic flexibility as potential mechanisms to mitigate liver injury [10]. Furthermore, moderate- to high-intensity aerobic, resistance, and combined exercise have been demonstrated to reduce liver fat and improve liver enzymes and HOMA-IR [19].” (lines 65 to 74, Introduction section).
In the discussion section, the authors should clarify the potential limitations of the study, particularly in only recruiting overweight/obese adolescents. The influence of hormones produced during puberty should also be considered.
Answer: Thank you for your interesting comment. We agree with the expert reviewer and more information has been added in the revised manuscript as follows “Third, the study involved overweight/obese adolescents, and therefore it is difficult to generalize the findings to normal-weight subjects of different ages. Additionally, we did not assess hormones produced during puberty that may influence our results. Puberty is associated with hormonal changes, mainly the release of sex hormones (estradi-ol/progesterone). Also, prolactin, growth hormone, insulin, and insulin-like growth factor-1 levels rise noticeably. These changes are responsible for fat mass gains that alter aerobic capacity and cardiometabolic parameters by increasing body size, and dimen-sion of the heart, lungs, muscles, and circulatory system [61, 62]. (lines 326 to 334, Discussion section).
Could high-intensity interval training have practical applications and be included in school curricula or clinical settings for adolescents with obesity or overweight status?
Answer: We agree with the expert reviewer's comment, and thus more information has been added in the revised manuscript “HIIT might be integrated into school curricula and clinical settings for overweight/obese adolescents to reduce body mass excess and liver fat, improve physical fitness, and promote cardiometabolic health. This would prevent/reverse MASLD and avoid its awful complications. Further research is needed to clarify the mechanisms underlying the effects of HIIT on fatty liver while focusing on the potential direct action of muscle metabolism on the liver.” (lines 317 to 322, Discussion section).

Round 2
Reviewer 2 Report
Comments and Suggestions for Authors
My compliments to the authors for addressing all the comments raised. The quality of the manuscript has improved significantly, and I therefore consider it suitable for publication in Nutrients.
Author Response
Manuscript ID: nutrients-3346290
Title: High-intensity interval training reduces liver enzyme levels and improves NAFLD-related biomarkers in overweight/obese girls
Response to the Academic Editor
We feel great thanks for your review of our article. We are very grateful to receive your comments which are very helpful for improving our paper. We have carefully considered the comments and revised the manuscript thoroughly. Detailed point-by-point is given below. Revised portions are highlighted in red characters in the revised manuscript.
Academic Editor Notes
In this study, the authors prove that exercise therapy alone can result in weight loss. To prevent MASLD, not only exercise therapy but also improvements in diet and physical habits should be implemented at the same time.
Answer: Thank you for your interesting comment. The discussion was revised according to your recommendation. The following paragraph has been added in the discussion section ‘However, to prevent MASLD, not only exercise therapy but also improvements in diet and physical habits should be implemented at the same time.’ (Discussion section, page 10).
This research is valuable and important, but there is insufficient discussion. 1. Please add a discussion about why the subjects in this study had a BMI of 30 or above. 2. What percentage of teenagers in Kef have a BMI over 30? 3. Are there many teenagers with high BMIs due to economic reasons, such as the average income of Kef residents? Is the reason why there are so many teenagers with a BMI of over 30 in Kef due to the lifestyle habits, such as diet, of Kef residents?
Answer: We appreciate your suggestion. Considering the Editor’s suggestion, we have reviewed the literature on Tunisian adolescents, but not specifically from the Kef region. To make the manuscript more comprehensive, we added data on the prevalence and reasons for increased adolescent obesity in Tunisia in the revised manuscript. The following paragraph has been added in the discussion section ‘Childhood obesity has become a prominent public health concern in transitional countries like Tunisia. The country experienced socioeconomic and nutritional transition and urbanization. Traditional Mediterranean diets have been supplanted by high-energy, fatty, and salty diets [29,30]. Concurrently, children and adolescents have become more engaged in sedentary activities such as surfing the Internet and playing with electronic devices, drastically reducing physical activity [30,31]. The prevalence of overweight among Tunisian adolescents rose from 10.8% in 1996 to 18.9% in 2005, then 30.5% in 2016 [32]. A recent study involving 1399 high-school Tunisian students showed that 20.4% are overweight and 7% are obese [33].’
